# SUMOylation and Viral Infections of the Brain

**DOI:** 10.3390/pathogens11070818

**Published:** 2022-07-21

**Authors:** Fergan Imbert, Gabrielle Leavitt, Dianne Langford

**Affiliations:** Department of Neural Sciences, Lewis Katz School of Medicine, Temple University, Philadelphia, PA 19140, USA; fergan.imbert@temple.edu (F.I.); gabrielle.leavitt@temple.edu (G.L.)

**Keywords:** SUMOylation, post-translational modifications, brain, neuroinflammation, HIV, ZIKA, cytomegalovirus, microglia, coronavirus

## Abstract

The small ubiquitin-like modifier (SUMO) system regulates numerous biological processes, including protein localization, stability and/or activity, transcription, and DNA repair. SUMO also plays critical roles in innate immunity and antiviral defense by mediating interferon (IFN) synthesis and signaling, as well as the expression and function of IFN-stimulated gene products. Viruses including human immunodeficiency virus-1, Zika virus, herpesviruses, and coronaviruses have evolved to exploit the host SUMOylation system to counteract the antiviral activities of SUMO proteins and to modify their own proteins for viral persistence and pathogenesis. Understanding the exploitation of SUMO is necessary for the development of effective antiviral therapies. This review summarizes the interplay between viruses and the host SUMOylation system, with a special emphasis on viruses with neuro-invasive properties that have pathogenic consequences on the central nervous system.

## 1. Introduction

While the central nervous system (CNS) has historically been viewed as an immune-privileged site, it is now appreciated that a robust immune response occurs during pathogenic or mechanical challenge [1,2]. Specialized resident CNS immune cells are responsible for surveillance during steady-state conditions and are the first to respond to foreign pathogens and/or cellular or tissue damage. However, viral infections can disrupt some of these cell signaling pathways to promote viral replication, host cell survival, or immune evasion. For example, many viruses have evolved mechanisms to regulate the activity of the host post-translational small ubiquitin-like modifier (SUMO)ylation system to sustain viral infection even in the presence of immune surveillance of the brain [3,4,5,6,7,8]. In this respect, SUMOylation manipulation represents an important target for persistent viral infection, but also a potential therapeutic target for managing or eliminating viral diseases. For example, a better understanding of the potential role(s) of SUMOylation in chronic human immunodeficiency virus-1 (HIV-1) infection of the brain and persistent viral latency may lead to novel therapeutic strategies and will provide insight into the molecular and cellular consequences of SUMO activity in healthy and infected cells. Here, we provide a review of current studies that summarize the involvement of SUMOylation during viral infections of the brain.

## 2. SUMOylation

The post-translational modification of proteins by SUMOs plays a significant role in regulating the host proteome. In general, SUMOylation may affect protein localization, stability, and/or activity by altering the dynamics of protein–protein interactions [9]. The human genome encodes four SUMO proteins, SUMOs 1–4, all of which are ~10 kD in size. SUMO2 and SUMO3 (commonly referred to as SUMO2/3) share significantly more sequence similarity (96%) to one another than to either SUMO1 or SUMO4, and this is reflected in the specificity of SUMOylated targets, as SUMO1 and SUMO2/3 can have distinct targets from one another [10]. SUMO4 shares 87% identity with SUMO2/3, but its function is unclear, given that it does not undergo maturation by SUMO proteases under normal conditions [11,12,13]. SUMO proteins are expressed as precursors that require C-terminal proteolytic processing by SUMO-specific proteases (SENP) to expose a diglycine motif that is essential for conjugation to the target protein [14]. SUMO4 lacks this diglycine residue and likely does not undergo processing and conjugation under normal conditions, acting only in a stress-dependent manner [15,16]. There are six mammalian SENP family members (SENP1-3 and SENP5-7), each with distinct substrate specificity and tissue distribution [17].

Like ubiquitination, the conjugation of mature SUMO to protein substrates is mediated by a pathway consisting of E1, E2, and E3 enzymes (Figure 1). However, unlike ubiquitination, SUMOylation relies on a single E2-conjugating enzyme, Ubc9, which is highly conserved from yeast to humans. SUMOylation is a dynamic and reversible process and the removal of SUMO from conjugated proteins is mediated by the SENP family of proteins. SUMO can be attached to substrates as a single SUMO moiety (mono-SUMOylation), or at multiple lysine residues (multi-SUMOylation). In the case of SUMO2/3, SUMOylation can give rise to polymeric chains, whereby SUMO–SUMO linkages occur at ΩKXE sequences in their N-terminal extensions (poly-SUMOylation) [9,18]. These interactions are mediated by the recruitment of binding partners that contain SUMO-interacting motifs (SIMs), which are characterized by a stretch of acidic and/or serine residues and a hydrophobic core [19,20]. In some instances, SUMO chains are recognized by SUMO-targeted ubiquitin ligases, or STUbLs, that catalyze the addition of ubiquitin to SUMOylated proteins. This STUbL activity results in proteins that are modified by both ubiquitin and SUMO, thereby targeting them for proteasomal degradation. SUMOylation plays a significant role in processes such as signal transduction, epigenetic modifications, and DNA repair [21,22,23]. Moreover, the dysregulation of SUMOylation has been shown to be associated with various diseases, including neurodegenerative diseases and some cancers [24,25].

## 3. SUMO Responses to Viral Infections

### 3.1. SUMO and HIV

HIV-1 infection is marked by the progressive depletion of peripheral CD4^+^ T cells and is the causative factor of acquired immunodeficiency syndrome (AIDS). People with HIV (PWH) that take effective combination anti-retroviral therapy (cART) can live normal life spans but may suffer from age-related disorders at an earlier age than uninfected individuals. Despite the success of cART, HIV infection remains a major health issue worldwide, and eradicating the virus from latent reservoirs is a significant barrier to a functional cure. Not only can latent reservoirs in the CNS provide a source of new viral particles capable of replenishing viral loads in the periphery, but persistent HIV infection can also lead to the development of HIV-associated neurocognitive disorders (HAND) [27,28]. HAND is fueled in part by the immune activation of macrophages and microglia [29]. More severe forms of HAND, like HIV-associated dementia (HAD), are far less prevalent in people with HIV (PWH) on cART, but asymptomatic neurocognitive impairment (ANI) and mild neurocognitive disorder (MND) have a prevalence of up to 50% in PWH, regardless of ART status [30].

In the CNS, microglia are the main cell type infected by HIV, and are also the reservoir for latent virus in the brain [31]. Recent evidence reports that astrocytes are also susceptible to HIV-1 infection and can release viral particles that may egress from the brain to the periphery [32,33], but their contribution to supporting latent virus is controversial. However, research using several methods, including transfection with proviral DNA, transduction with vesicular stomatitis virus (VSV)-G pseudotyped viruses, or transient expression of CD4 followed by HIV infection, demonstrated that persistent—but largely non-replicating—infection could be established in astrocytes [32]. Notably, unlike HIV infection of CD4+ T-cells that results in cell lysis, the infection of microglia is non-lytic, thereby allowing for the persistence of HIV-1 in the brain [34]. In this case, viral replication is not essential, but viral DNA persists in the host genome. In HIV-infected microglia, host transcription factors promote the establishment and persistence of latency to prevent viral replication. Coup-TF-interacting protein-2 (CTIP2) facilitates HIV-1 latency through the formation of heterochromatin at the viral promoter that leads to HIV-1 silencing (Figure 2) [35]. CTIP2 serves as the anchor for a chromatin remodeling complex (or viral latency complex) consisting of several transcription factors, histone deacetylases/methyltransferases, and ubiquitin ligases [35]. Further investigation of the molecular mechanisms of HIV-1 latency in glial cells can help identify new targets to achieve a functional cure.

Like DNA and RNA viruses, HIV has evolved multifaceted measures to evade host immune responses and achieve productive infection that include taking advantage of the host cellular SUMOylation machinery. HIV can achieve productive infection in host cells by modifying its viral proteins or redirecting essential ligases of the SUMO pathway to regulate global cellular SUMOylation levels (reviewed in: [26,36]. Several studies highlight the role of SUMO in HIV-infected cells. For example, the SUMOylation of HIV-1 integrase (IN) abrogates proper HIV function. While SUMOylation has no effect on HIV’s ability to infect the host cell, the SUMOylation of HIV IN renders virus replication deficient [37]. Moreover, the ubiquitination of the p6 domain of the HIV-1 Gag polyprotein is important for the recruitment of cellular factors that mediate the trafficking of endocytic vesicles and virion release, but the interaction of p6 with SUMO-1 blocks its ubiquitination [38]. Interestingly, the overexpression of SUMO-1 does not reduce virion release, but instead decreases the infectivity of the released virions in HEK293 cells [38]. Together, these in vitro studies suggest that SUMOylation may counteract HIV in the brain by targeting both replication and virion infectivity.

As mentioned above, there are several transcription factors and ubiquitin ligases involved in the establishment and persistence of latency in microglia. Importantly, several of these players are SUMO substrates, SUMO E3 ligases, or SIMS. The transcriptional regulator, CTIP2, is responsible for recruiting the multi-enzyme chromatin modifying complex at the HIV-1 promoter and contains two SUMOylation sites (Lys-679 and Lys-877) [39]. CTIP2 also associates and cooperates with the histone methyltransferase, Suv39h1, to repress HIV-1 gene transcription [35]. Suv39h1 is a SUMO1-interacting protein that directly interacts with Ubc9, a characteristic of SUMO E3 ligases [40]. Notably, Suv39h1 can promote heterochromatin protein-1 (HP1a) SUMOylation in vivo, which is consistent with SUMO E3 ligase activity [40]. Similarly, more recent work demonstrated that tripartite motif containing-28 (TRIM28) (also known as KAP1) cooperates with CTIP2 to repress HIV-1 gene transcription in microglial cells. TRIM28 acts as a SUMO E3 ligase for itself, and for other proteins including the interferon regulatory factor-7 (IRF7), vacuolar protein sorting-34 (Vps34), and the DNA replication factor proliferating cell nuclear antigen (PCNA) [41,42,43,44]. Whether TRIM28 utilizes SUMO E3 ligase activity in cooperation with CTIP2 to potently suppress HIV-1 expression remains unclear. The transcription factor specificity protein-1 (Sp1) is responsible for the recruitment of CTIP2 to the viral promoter and is differentially regulated by SUMOs 1 and 2 [45,46]. Mechanistically, SUMO1 promotes interactions between Sp1 and the histone acetyltransferase, p300, while SUMO2 interferes with this interaction and decreases Sp1 protein stability [45]. Interestingly, SUMOylation has not been assessed in the establishment and/or persistence of HIV latency in microglia. A deeper understanding of how SUMOylation regulates both viral proteins and cellular antiviral components in the context of latency in microglia can lead to the development of effective antiviral therapies.

### 3.2. SUMO and Coronaviruses

Coronaviruses (CoVs) are a diverse family of enveloped positive-sense single-stranded RNA viruses that can infect a wide variety of avian species and mammals, including humans. Human coronaviruses circulate in the population and cause seasonal, mild respiratory infections. Conversely, emerging Middle East respiratory syndrome coronavirus (MERS-CoV) and severe acute respiratory coronavirus (SARS-CoV) are highly pathogenic and can develop into life-threatening respiratory diseases [47,48,49]. SARS-CoV-2, which is responsible for the ongoing pandemic, marked the third introduction of a highly-pathogenic coronavirus into the human population [50,51]. Although there is still a great deal that we do not understand about SARS-CoV-2, our knowledge of viral and host factors that contribute to COVID-19, the disease induced by SARS-CoV-2 infection, is accumulating at an unprecedented rate. Mechanistically, SARS-CoV-2 infection of a human host cell involves multiple, coordinated processes, including viral protein (S-gp) binding to host receptors, angiotensin-converting enzyme-2 (ACE2) and transmembrane serine protease-2 (TMPRSS2); and the triggering of host immunological and inflammatory responses (also known as a cytokine storm) [52,53,54]. Cytokine storms associated with COVID-19 appear to be a leading cause of mortality [55].

The neuro-invasive potential of coronaviruses in both human and other animal brains is documented [56,57]. Given the similarity between SARS-CoV and SARS-CoV-2, it was hypothesized that SARS-CoV-2 enters the CNS. In fact, increasing evidence indicates that SARS-CoV-2 uses several mechanisms to enter the CNS and that viral interaction with the cardiorespiratory brainstem center is a contributing factor to the death of infected mice and humans [58,59]. While the consequences of SARS-CoV-2 on cells of the CNS remain unclear, a major focus has been placed on understanding the interaction between SARS-CoV-2 and glial cells. Glial cells, including astrocytes and microglia, are critical for maintaining the integrity of the blood-brain barrier (BBB) and overall brain health, so understanding the relationship between SARS-CoV-2 and glial cells will offer insight into the local and systemic pathogenesis of COVID-19. There is evidence of reactive astrogliosis in COVID-19, accompanied by a significant increase in the plasma levels of the astrocyte-specific protein, glial fibrillary acidic protein (GFAP), in patients with moderate to severe COVID-19 [60]. Much less is known about microglia in SARS-CoV-2 but considering that microglia are highly motile and are in a constant “immunologically alert” state, they may contribute to the neurological complications and neuro-inflammation observed in COVID-19 patients [61] (Figure 3). For example, in Figure 3 from Thakur et al., microglial nodules are apparent and surround neurons with aberrant morphology associated with degeneration [61].

While protein SUMOylation has been recognized as a critical component of viral pathogenesis, little is known about the relationship between SUMOylation and coronaviruses. A limited number of studies have identified an interaction with the SUMO machinery and the nucleocapsid (N) protein of SARS-CoV. The coronavirus N protein is a multifunctional protein essential for proper nucleocapsid assembly and genomic RNA replication [62]. Importantly, the SUMO E2 enzyme, Ubc9, was identified as an interacting partner of the N protein [63,64]. Biochemical analyses also revealed that SUMO1 modification at K62 of the N protein induces homo-oligomerization [64], which is required for the formation of the viral capsid to protect the viral genome. Given that homo-oligomerization of the N protein is essential for a stable conformation, SUMOylation may play a key role in the SARS-CoV replication cycle. Whether the N protein of SARS-CoV-2 is also SUMOylated is not known; however, given the significant sequence similarity of the N protein among the coronaviruses, it is possible that the SARS-CoV-2 N protein is modified by SUMO as well [65,66].

### 3.3. SUMO and Zika Virus

Zika virus (ZIKV) is a positive-sense single-stranded RNA flavivirus that is transmitted predominantly by mosquitoes. However, sexual and maternal–fetal transmission have been reported as other mechanisms of transmission [67]. Although the majority of ZIKV infections are asymptomatic, symptomatic ZIKV infections manifest through joint and muscle pain, as well as a rash and low-grade fever [67]. More severe neurological manifestations such as hydrocephalus and ZIKV-associated Guillain–Barré syndrome (GBS) may occur in some infected individuals, both of which are associated with high morbidity [68]. GBS is an autoimmune disease that attacks the peripheral nerves, and hydrocephalus is a well-documented, but rare, complication in individuals with the disorder [69,70]. Hydrocephalus has also been associated with congenital Zika syndrome, a group of birth defects associated with prenatal exposure to ZIKV [71]. ZIKV has also been shown to infect and replicate in mature neurons and glial cells, induce neuroinflammatory processes, and has been linked to myelitis, peripheral neuropathy, and reduced gray matter volume in motor-associated cortical regions [72,73]. However, the exact mechanisms underlying ZIKV-induced neurological disorders have yet to be fully elucidated.

SUMO proteins have been shown to interact with ZIKV, particularly the non-structural-5 (NS5) viral protein [74]. NS5 encodes both viral methyltransferase and RNA-dependent RNA polymerase. This protein is important in viral replication and to ensure viral survival and replication by inhibiting the host’s innate immune system [75]. Despite flaviviruses replicating in the cytoplasm, NS5 proteins of ZIKV are predominantly located in the nuclei of infected cells [74]. Multiple sequence alignments of over 400 pre-epidemic and epidemic ZIKV strains revealed a putative SUMO-interacting motif (SIM) at the N-terminal domain of NS5 [74]. NS5 is stabilized by SUMOylation and is critical for the persistent ZIKV infection of human brain microvascular endothelial cells (hBMECs) [76]. Specifically, during ZIKV infection, SUMOylation of NS5 decreases its ubiquitin-mediated degradation and promotes its interaction with the signal transducer and activator of transcription-2 (STAT2) protein, which disrupts the host antiviral promyelocytic leukemia (PML)-STAT2 nuclear bodies (NBs) and leads to the degradation of PML [76]. In fact, as previously reported by Conde et al., NS5 forms SUMO-1-co-localized and SUMO-1-independent nuclear speckles during ZIKV replication in human brain microvascular endothelial cells (Figure 4). Following ZIKV infection, distinct nuclear speckles of NS5 were observed in the nucleus of hBMECs that co-localized with SUMO1 (Figure 4A). Importantly, ZIKV infection reduced the association of SUMO1 with PML and reduced PML expression levels (Figure 4B). Given that PML promotes the transcription of interferon-stimulated genes (ISGs), PML degradation leads to a disruption in type-I interferon signaling, fostering a favorable environment for viral persistence and pathogenesis [77].

Recent studies exploring therapeutic strategies for the treatment of flavivirus infections showed that the SUMO inhibitor, 2-D08, significantly reduced ZIKV replication and protected cells from ZIKV-induced cytotoxicity in vitro [74]. The same study also reported that SIM-mutated ZIKV NS5 failed to suppress type-I interferon signaling [74]. In studies with acute myeloid leukemia cells, 2-D08 induced apoptosis by de-SUMOylating the NAPDH oxidase 2 (NOX2), thereby activating NOX2-mediated ROS production [78]. Although NOX2 signaling in ZIKV has not been described, similar mechanisms may be involved in the 2-D08 inhibition of ZIKV replication, since binding between the SUMO1 protein and the ZIKV putative NS5 SIM peptide is predicted (Figure 5). The amino acid sequence of the putative SIM of NS5 is conserved between flaviviruses (Figure 5A) and a molecular docking model revealed that the VIDL segment (SIM core sequence) of NS5 forms interactions with the active site of SUMO1 (Figure 5B,C). Together, these studies suggest that flaviviruses have an evolutionarily conserved mechanism that enhances virus proliferation while suppressing host antiviral responses through SUMO modification of the viral NS5 protein. The findings from these studies highlight potential therapeutic uses for SUMO inhibitors along with antiviral treatments for infections caused by ZIKV or other flaviviruses.

### 3.4. SUMO and HCMV

Human cytomegalovirus (HCMV) is a beta-herpesvirus that causes lifelong infection in humans. An infected person can spread viral particles through bodily fluids such as saliva, urine, blood, tears, semen, and breast milk. It has an incredibly high prevalence of over 75% in adults, however, for many, the infection remains latent [79]. Primary infection in healthy adults can include symptoms such as fever, fatigue, sore throat, and swollen lymph nodes. HCMV can also infrequently lead to mononucleosis or hepatitis. HCMV is an opportunistic pathogen and HCMV symptoms may become more severe in people with weakened immune systems [80]. Several antiviral medications including ganciclovir, valganciclovir, foscarnet, and cidofovir are often prescribed for treatment, but success is extremely poor with recurrence usually within a year; the emergence of drug-resistant HCMV has been reported for all antivirals listed [81].

Studies of HCMV infection in the brain are currently limited due to the absence of an animal model that accurately mimics human infection. Murine CMV (MCMV) has considerable sequence homology to HCMV, so mouse models are often used to study congenital HCMV infection [82]. However, the neuro-invasive potential of MCMV is dependent on immune deficiency [83]. Thus, what is currently known of the pathogenesis of HCMV in the CNS is based largely on clinical features and post-mortem studies. The histopathological analysis of brains from infants with severe congenital HCMV infection suggested that ependymal cells and neural stem/progenitor cells (NSPC), a neuroepithelial precursor, are prime targets for HCMV [84]. While HCMV also infects neurons and oligodendrocytes, astrocytes are the primary target [85,86]. There is currently no evidence that microglia are productively infected with HCMV or that there are any major cytopathic presentations [85,87]. HCMV eventually becomes latent in infected T-cells, lymphocytes, and macrophages [88].

Notably, the HCMV immediate–early-1 (IE1) protein was the first viral protein to be identified as a SUMO substrate (Figure 6) [89]. However, the interaction between the host SUMOylation system and other HCMV proteins remained largely unknown until a group described the role of HCMV viral protein, pp71, in the SUMOylation of the host protein, Daxx [90]. HCMV pp71 is delivered immediately upon infection of host cells by HCMV virions and promotes the SUMOylation of its cellular substrate, Daxx, though the role that this modification plays in regulating Daxx activity is unknown. The DNA polymerase processivity factor, UL44, was later identified as a binding partner of the SUMO-conjugating enzyme, Ubc9 [91]. Data indicated that the overexpression of SUMO altered the intranuclear localization of UL44 in HCMV-infected cells, resulting in increased viral replication [91]. However, a more recent study reported that the removal of lysine 410 (K410) within the SUMO consensus motif located in the C-terminus of UL44 enhanced viral DNA synthesis and viral production in HCMV-infected cells [92]. Further analyses of the interactions between HCMV proteins and SUMOylation is necessary to reconcile these contradictory experimental observations. The latency-associated gene product (LUNA) was recently revealed to encode a conserved deSUMOylase motif (Asp-Cys-Gly), which is responsible for promoting PML deSUMOylation and priming the cell for viral reactivation [93]. These studies illustrate the role for the host SUMOylation system during HCMV infections and the importance of understanding the role of SUMOylation in the innate immune response.

### 3.5. SUMO and Herpes Simplex Virus

Herpes simplex virus, commonly referred to as herpes, is a linear dsDNA enveloped virus. Herpes is divided into two types: HSV-1 and HSV-2. HSV-1 is typically transmitted by oral-to-oral contact and usually causes an infection in or around the mouth (oral herpes). HSV-2 is mainly sexually transmitted and leads to genital herpes. Neurons, epithelial cells, and keratinocytes are targets for HSV infection [94]. Both oral and genital herpes range in symptom severity—from asymptomatic to painful ulcers and blisters at the site of infection. HSV is a lifelong disease once infected, with periods of latency (HSV-1 in the trigeminal ganglia and HSV-2 in the sacral nerve root ganglia) and reactivation [95]. Antiviral medications such as acyclovir, famciclovir, and valacyclovir are the most effective treatment for people with HSV. These antivirals work to reduce the severity and frequency of symptom flare-ups. In immunocompromised patients, HSV can cause more severe symptoms. For example, complications of HSV1/2 and HIV-1 co-infection include encephalitis, keratitis, and disseminated infection [96].

HSV can utilize multiple mechanisms for host cell entry. In some instances, HSV attachment and fusion with host cells is facilitated by the association of the viral glycoprotein(s) (gB, gC, gD, and/or the gH/gL complex) and herpes simplex proteoglycans (HSPGs) present on the cell surface [97]. Similarly, Nectin-1, a calcium independent immunoglobulin cell–cell adhesion molecule, interacts with HSV gD to allow for HSV entry into host cells. Nectin-1 is implicated in the development of conjunctivitis and epithelial keratitis due to HSV-1 infection and in acute retinal necrosis due to both viral subtypes [98]. Herpes virus entry mediator (HVEM), a member of the tumor necrosis factor receptor superfamily involved in inflammatory regulation, also interacts with HSV gD to promote cell entry [99]. A modified form of heparan sulfate (3-OS HS) produces a receptor that can bind HSV gD, thereby providing another mechanism of entry and is implicated in stromal keratitis and conjunctivitis [100]. Recent studies have shown that the co-receptor immunoglobulin-like-2 (PILR) α, associates with gB and is involved in HSV-1 fusion to host cells, however the importance of this interaction is unknown [101]. Filopodia, which are thin actin-rich plasma membrane protrusions, are also implicated in viral spread from cell to cell [102]. HSV-1 can also use endocytosis as a mechanism of cellular entry. The standard clatherin-coated pit-mediated endocytosis has been observed as well as a more unique pH-dependent phagocytosis-like mechanism in which protrusions of the plasma membranes with no clatherin coats engulfed the enveloped virions [103].

HSV-1 establishes latency in both peripheral nerve ganglia and the central nervous system (CNS). In rare cases, HSV-1 can replicate in the CNS, triggering an acute infection and inflammatory response leading to herpes simplex encephalitis (HSE). In vivo experimental models have identified microglial cells as the source of pro-inflammatory cytokines and chemokines in response to HSV-1, through the recruitment of circulating lymphocytes [104]. HSV-1-infected microglial cells also produce reactive oxygen species (ROS) that exacerbate disease progression [105].

Like the other viruses discussed in this review, SUMOylation and SUMO-mediated interactions play important roles in HSV-1 infection. Similar to ZIKV, certain components of PML nuclear bodies have been identified as major contributors to intrinsic resistance to viral infections. PML nuclear bodies are heavily modified by the SUMO proteins and are key factors for their assembly [106]. The HSV-1 regulatory protein-infected cell polypeptide-0 (ICP0) counteracts the intrinsic anti-HSV properties of PML through its ubiquitin E3 ligase activity [107,108]. ICP0 not only targets PML more efficiently than SUMO, but induces PML degradation through SUMO-targeted ubiquitin ligase (STUbL)-like activities [5,108]. An analysis of the SUMO2 sub-proteome in HeparRG hepatocytes revealed several proteins whose unSUMOylated forms are also degraded during HSV-1 infection, though whether these are also substrates of ICP0 remain unclear [109].

## 4. Conclusions

The interplay among post-translational modifications like SUMOylation and CNS viral infections is multifaceted (Table 1). Recent studies have underscored the importance of SUMO proteins in antiviral responses and how these same proteins can be manipulated by viruses to achieve viral replication and persistence. Several viruses have evolved to exploit SUMOylation processes of the host cell either by targeting different steps of the SUMOylation system (i.e., the SUMO enzymes) or through covalent modification of specific viral and cellular proteins (Table 1). As outlined here, several of these viruses have neuro-invasive properties. For example, inhibiting the SUMOylation of the SARS-CoV NP, which functions to encapsulate the viral genome, could attenuate virus propagation. Similarly, inhibiting SUMO1 conjugation of the ZIKV NS5 protein could drastically limit the survival and infectivity of ZIKV. Given that some viral infections in the CNS can be fatal or result in a lasting deficit, due in part to immune-mediated inflammation in the brain, an increased understanding of SUMOylation as a potential therapeutic target for viral infections of the CNS is warranted. Elucidating how viruses exploit the host’s SUMOylation pathway may reveal new targets for antiviral therapies.

## Figures and Tables

**Figure 1 pathogens-11-00818-f001:**
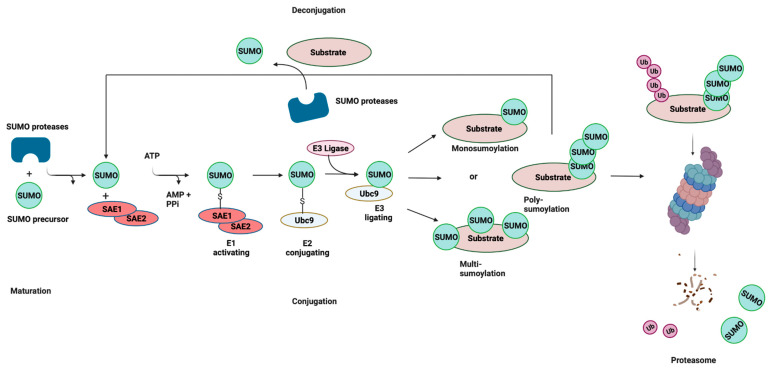
SUMOylation: the Small Ubiquitin-related Modifier (SUMO) pathway. SUMO proteins are processed by a SUMO-specific protease at the C-terminal tail to expose a diglycine (-GG) motif, resulting in a mature SUMO peptide. SUMO is subsequently activated in an ATP-dependent reaction, creating an intermediate thioester bond between the active site of SUMO and the heterodimeric E1-activating enzyme (SAE1/SAE2). Following activation, SUMO is transferred from the E1 enzyme to Ubc9, and finally attached to a target lysine in the protein substrate, which is usually located within the consensus site. This final step is mediated by E3 ligase enzymes that function in substrate recognition and specificity. SUMOylation is reversible (deconjugation), and the same SUMO proteases involved in SUMO maturation catalyze the removal of SUMO from the target substrate. Cross-talk exists between SUMOylation and ubiquitylation through SUMO-targeted ubiquitin ligases (STUbLs). STUbLs are enzymes that catalyze the addition of ubiquitin to proteins that have been previously SUMOylated with SUMO chains. STUbL activity results in target proteins that are modified by both SUMO and ubiquitin, which can be targeted to the proteasome for degradation. AMP: adenosine monophosphate; PPi: pyrophosphate; Ub: ubiquitin. Adapted with permission from [26] 2021, Springer Nature. Figure was created with Biorender.

**Figure 2 pathogens-11-00818-f002:**
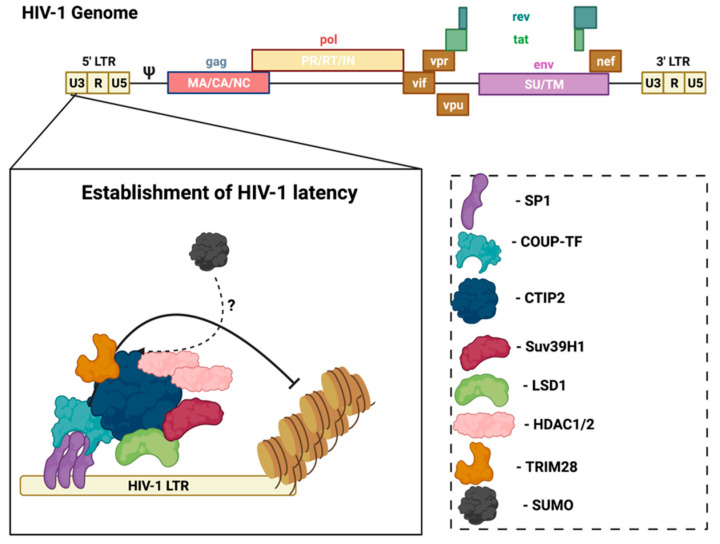
CTIP2 promotes the establishment of HIV-1 latency in microglia. CTIP2 participates in the establishment of HIV-1 latency in microglia by recruiting a chromatin modifying complex (or viral latency complex) to the viral promoter (HIV-1 LTR). This complex consists of: Sp1, which anchors CTIP2 to the viral promoter and acts as a scaffold for the recruitment of chromatin modifying proteins; HDAC1 and HDAC2, which are responsible for deacetylation of Nuc-1, one of the nucleosomes positioned immediately downstream of the transcriptional start site of the HIV-1 LTR; and the histone methyltransferase Suv39H1, which contributes to HIV-1 silencing through methylation of Nuc-1. CTIP2 also recruits the demethylase, LSD1, and the SUMO E3 ligase, TRIM28, which, in association with CTIP2, contributes to HIV-1 gene silencing. Several of the CTIP2-associated proteins in the viral latency complex interact with the host SUMOylation system. Accordingly, determining the role of the SUMOylation in the establishment and/or persistence of HIV-1 latency in microglia could aid in the design of new pharmacological agents that target HIV-1 viral reservoirs. Sp1: specificity protein 1; COUP-TF: chicken ovalbumin upstream promoter transcription factor; CTIP2: COUP-TF interacting protein 2; HDAC1/2: histone deacetylase 1/2; TRIM28: tripartite motif containing 28; SUMO: small ubiquitin-related modifier. Figure was created with Biorender.

**Figure 3 pathogens-11-00818-f003:**
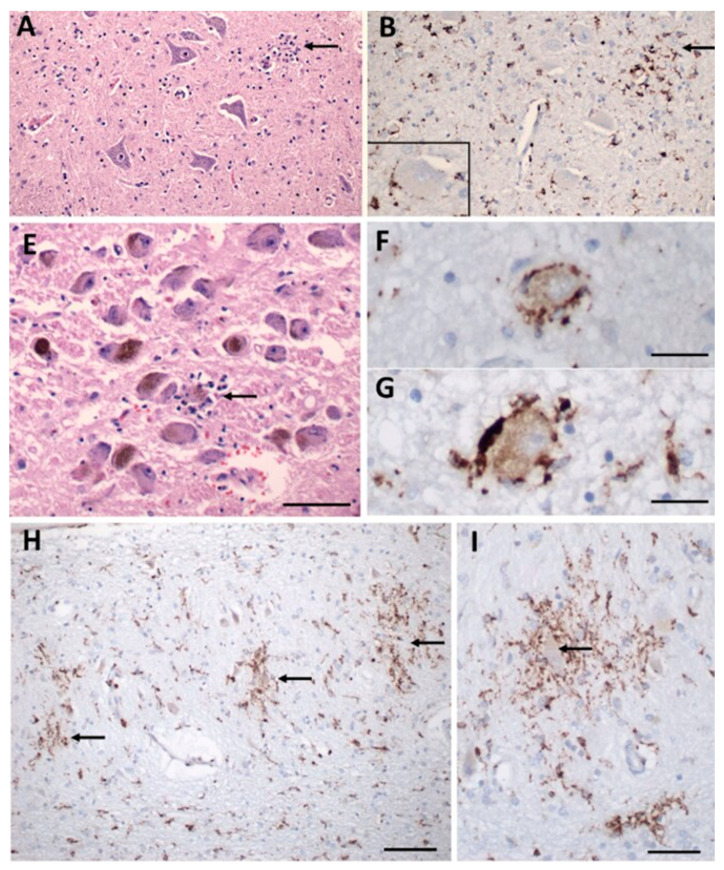
Inflammatory pathology in COVID-19 brains. (**A**) Section of the hypoglossal nucleus shows several motor neurons and a microglial nodule (arrow). (**B**) An adjacent section immunolabeled for CD68 (brown) shows clustered microglia within the nodule. Inset: Microglia in close apposition to a hypoglossal neuron (CD68^+^). (**E**) The locus coeruleus contains a microglial nodule with a degenerating neuron in the center, identified by its residual neuromelanin (arrow). (**F**,**G**) Neurons of the dorsal motor nucleus of the vagus surrounded by CD68^+^ microglia. (**H**,**I**) Microglial nodules in the dentate nucleus (arrows in (**H**)), neuron in the middle of a nodule (arrow in (**I**)), CD68. Scale bar in (**A**,**B**) = 200 µm; (**E**) = 10 µm; (**F**,**G**) = 50 µm; (**H**) = 100 µm; (**I**) = 50 µm. Adapted with permission from [61] 2021, Oxford University Press. Panels C, D, J, K from the original publication are not shown.

**Figure 4 pathogens-11-00818-f004:**
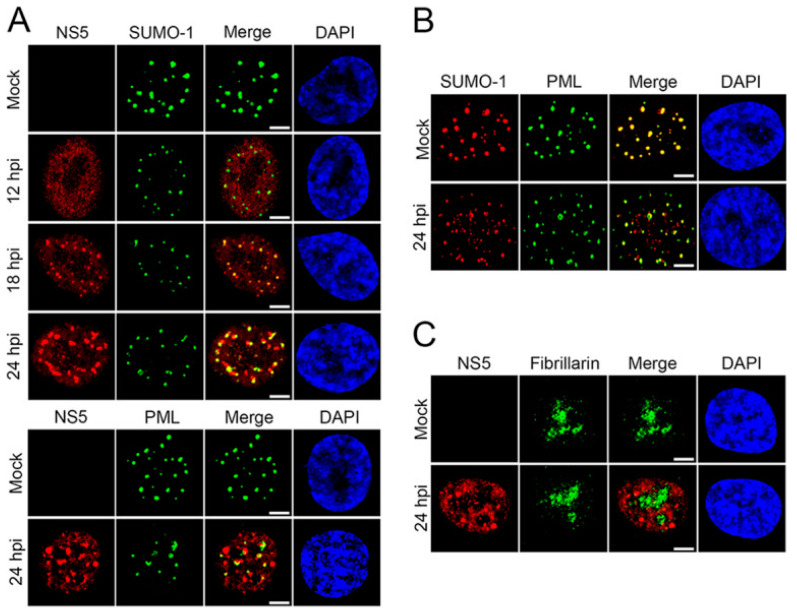
ZIKV NS5 forms unique nuclear speckles that disrupt PML/SUMO-1 NBs. (**A**) hBMECs grown on microslides were infected with ZIKV-PRV (MOI, 2) and fixed 12, 18, and 24 hpi; immunolabeled for ZIKV NS5 and SUMO-1 or PML; and visualized by confocal microscopy. (**B**) Mock-infected or ZIKV-infected hBMECs were immunolabeled for PML and SUMO-1 24 hpi and visualized by confocal microscopy. (**C**) ZIKV-infected hBMECs were immunolabeled for ZIKV NS5 and fibrillarin at 24 hpi and visualized by confocal microscopy. Experiments were conducted in triplicate, repeated at least three times, and representative data are presented. Bars = 5 μm. Adapted with permission from [76]. 2020, American Society for Microbiology. PML: promyelocytic leukemia; ZIKV-PRV: PRVABC59 ZIKV strain; hBMEC: human brain microvascular endothelial cells; MOI: multiplicity of infection; hpi: hours post-infection.

**Figure 5 pathogens-11-00818-f005:**
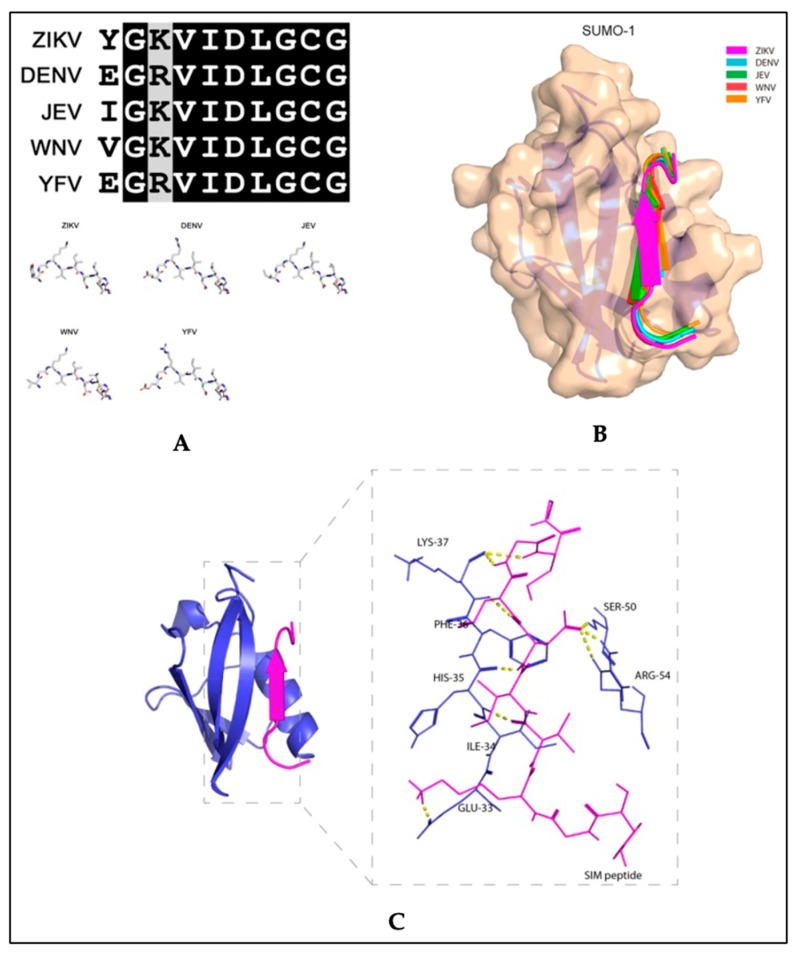
Molecular docking model of the binding between the putative non-structural 5 (NS5) SUMO-interacting motifs of flaviviruses and the SUMO1 protein. (**A**) Top panel: multiple sequence alignment of the amino acid sequences of the putative NS5 protein SUMO-interacting motifs (SIM) of Zika virus (ZIKV), dengue virus (DENV) (serotype 3), Japanese encephalitis virus (JEV), West Nile virus (WNV), and yellow fever virus (YFV). Bottom panel: Stick representation of the structural similarities among the five flaviviruses’ putative NS5 SIM peptides. (**B**) Schematic representation of the binding between the SUMO1 protein and the five flaviviruses’ putative NS5 SIM peptides. The SUMO1 protein is shown in tan and the NS5 SIM peptides are shown in different colors. (**C**) Ribbon representation showing the interacting amino acid residues of the putative ZIKV NS5 SIM peptide and the active sites of the SUMO1 protein. The putative ZIKV NS5 SIM peptide and SUMO1 protein are displayed in magenta and blue, respectively. The interacting residues are shown as sticks with hydrogen bonds represented by yellow dashed lines. Adapted with permission from [74]. 2019, MDPI.

**Figure 6 pathogens-11-00818-f006:**
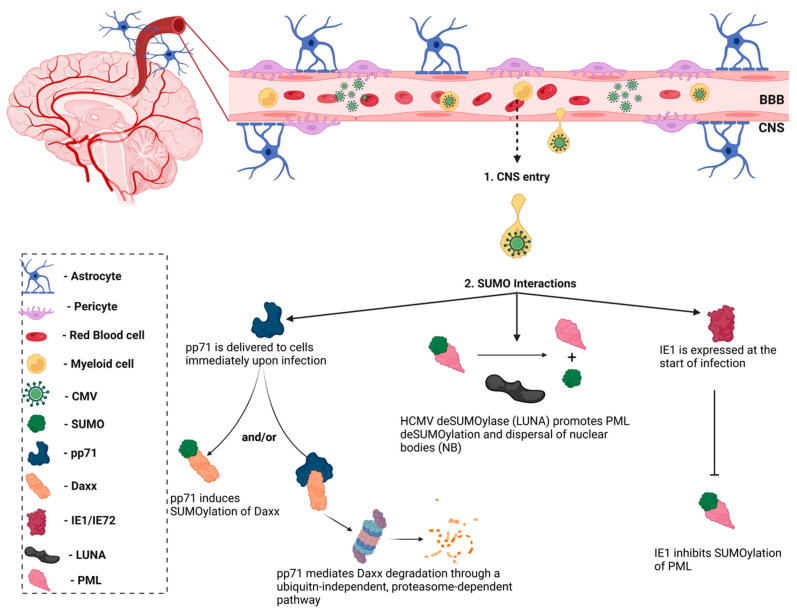
Interactions between human cytomegalovirus (HCMV) and SUMO. HCMV entry to the central nervous system (CNS) is secondary to peripheral organ infection. Passage across the blood-brain barrier is thought to be mediated by monocytes. Upon crossing the BBB, HCMV infects resident cells and has been shown to interact with the host SUMOylation system. Immediate–early protein-1 (IE1) was the first viral protein identified as a SUMO interactor and has been shown to inhibit the SUMOylation of promyelocytic leukemia (PML) bodies, which suppresses innate immune responses. Similarly, the HCMV latency-associated protein, LUNA, functions as a de-SUMOylase to promote PML de-SUMOylation. The HCMV pp71 protein promotes the SUMOylation of its cellular substrate, Daxx, though the functional consequence of this interaction is unknown. pp71 has also been shown to mediate Daxx degradation through a ubiquitin-independent pathway. Figure was created with Biorender.

**Table 1 pathogens-11-00818-t001:** Examples of cross-talk between neuro-invasive viruses and the host SUMOylation system.

Virus	Protein	SUMO Protein	SUMOylation Interactions	References
HIV	Integrase (IN)	SUMO1 and SUMO2/3	Inhibits viral genome integration	[37]
	P6	SUMO1	Reduces infectivity of released virions	[38]
SARS-CoV-1	NP	SUMO1	Promotes homo-oligomerization	[64]
HCMV	IE1	SUMO1	Abrogates interaction of SUMO1 with PML	[89]
	pp71	SUMO1	Promotes SUMOylation of Daxx	[90]
	UL44	SUMO1 and SUMO2/3	Enhances virus production and DNA replication	[91]
		SUMO1 and SUMO2/3	Attenuates virus production	[92]
	LUNA	N/A	DeSUMOylates PML	[93]
Zika	NS5	SUMO1	SUMO1 conjugation stabilizes NS5	[76]
HSV-1	ICP0	N/A	DeSUMOylates PML and induces it degradation	[108]

## Data Availability

Not applicable.

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
