# Peer review of "SUMOylation and Viral Infections of the Brain"

_pathogens, 2022, doi:10.3390/pathogens11070818_

Round 1

Reviewer 1 Report

The manuscript from Imbert et al., on SUMOylation and viral infections of the brain is a succinct and clear review of the subject.

I have only minor suggestions for the authors to improve the manuscript.

1. In Section 2 "SUMOylation" I suggest the authors to spell out why it is not clear whether SUMO4 is conjugated or not. For a reader that is not very familiar with the process of SUMOylation it would be helpful to explain first that the maturation and exposure of the diglycine residues is necessary for conjugation.

2. In the same section I would suggest the authors to revise their statement that SUMO 1 and SUMO2 have different targets, since it is well documented that they can share the same targets instead. The authors themselves later mention that the same protein is modified by both (CTIP2).

3. Finally in section 2 I believe the authors meant "proteome" and not proteasome, in the second line of section 2.

4. In section 3.1 on HIV infection the authors mention that SUMOylation counteracts HIV in the brain, but as far as I understand the authors only cites work performed on HEK293 cells. Maybe better say that the in vitro work suggests that SUMOylation may counteract the infection in the brain?

5. It would be useful to specify for each protein mentioned in the review whether there are evidence of SUMOylation by SUMO1 or SUMO2 or both. This could be easily added in the very useful Table that the authors already prepared.

Author Response

Thank you so much for your helpful comments. We have addressed each one as described below: 

  1. We have added a statement to clarify why SUMO4 is not processed and conjugated under normal conditions, by explaining that it lacks the diglycine motif.
  2. We have revised our statement that SUMO1 and SUMO2/3 have distinct targets and clarified that they can have distinct targets. 
  3. Thank you pointing out our error. We have corrected that sentence to state proteome. 
  4. We completely agree and have clarified that the in vitro work can have implication for infections in the brain. 
  5. This was a great idea and we have added it to the table. 

Reviewer 2 Report

Few original works from 2000 onward connected SUMO system with pathogenesis triggers by microorganisms. Bacteria and viruses are known to use SUMO system either by inhibiting global SUMOylation or exploiting components of SUMOylation pathway for their survivals inside host cells. Imbert et al., reviewed importance of SUMOylation in viral infections of the brain. In this work they have discussed various modes of SUMO-dependent regulation that are being employed by both the viruses and the host cells to counteracts each other for their biological success. A special section was devoted on the relationship of SUMOs with SARS-CoV2 type of coronavirus, providing many provocative thoughts on developing out-of-the-box strategies in controlling future pandemics involving highly contagious viruses such as corona virus. I think the review is well written, very contemporary and informative. This review will be of interest of a broad category of researchers and therefore, it deserves publication in Pathogen.

Author Response

Thank you for your positive comments. We hope that you enjoyed reading the review. 

Reviewer 3 Report

This is a very well written and thoughtful review about sumoylation and viruses infecting the CNS. The illustrations are  really good. I recommend this for publication.

Author Response

Thank you so much for your positive comments. I used Biorender to create the figures; it is quite a useful tool for academic illustrations. 

Reviewer 4 Report

This is a short review that presents the role of sumoylation in CNS infections by 4 viruses: HIV, Zika, coronaviruses, and cytomegalovirus. While the data on sumoylation and CNS infection are quite limited for most of these viruses, this review adequately summarizes the available literature. My only comment is about the absence of a section on the herpes simplex virus. As a neurotropic virus that has a large literature on its involvement with the sumoylation system, I was surprised that this virus wasn't addressed in the review. The authors should consider adding an additional section to discuss this virus.

Author Response

We completely agree. Herpes Simplex virus should have been included in this review and we have added the section. Thank you so much for your positive comments and we hope that you enjoyed reading the review.